# The Anticancer Drug Daunomycin Directly Affects Gene Expression and DNA Structure

**DOI:** 10.3390/ijms24076631

**Published:** 2023-04-01

**Authors:** Takashi Nishio, Yohji Shimada, Yuko Yoshikawa, Takahiro Kenmotsu, Helmut Schiessel, Kenichi Yoshikawa

**Affiliations:** 1Faculty of Life and Medical Sciences, Doshisha University, Kyoto 610-0394, Japan; takashinishio7329@gmail.com (T.N.); ydu.schalke402@gmail.com (Y.S.); yoshi2989r@gmail.com (Y.Y.); tkenmots@mail.doshisha.ac.jp (T.K.); 2Cluster of Excellence Physics of Life, TU Dresden, 01307 Dresden, Germany; helmut.schiessel@tu-dresden.de; 3Center for Integrative Medicine and Physics, Institute for Advanced Study, Kyoto University, Kyoto 606-8501, Japan

**Keywords:** anticancer drug, daunomycin, polyamine, higher-order structure of DNA, gene expression, double-strand breaks (DSBs), atomic force microscopy (AFM), cell-free in vitro luciferase assay

## Abstract

Daunomycin (DM), an anthracycline antibiotic, is frequently used to treat various cancers, but the direct effects of DM on gene expression and DNA structure are unclear. We used an in vitro cell-free system, optimized with spermine (SP), to study the effect of DM on gene expression. A bimodal effect of DM on gene expression, weak promotion followed by inhibition, was observed with increasing concentration of DM. We also performed atomic force microscopy observation to measure how DM affects the higher-order structure of DNA induced with SP. DM destroyed SP-induced flower-like conformations of DNA by generating double-strand breaks, and this destructive conformational change of DNA corresponded to the inhibitory effect on gene expression. Interestingly, the weakly enhanced cell-free gene expression occurred as DNA conformations were elongated or relaxed at lower DM concentrations. We expect these newly unveiled DM effects on gene expression and the higher-order structure of DNA will contribute further to the development and refinement of useful anticancer therapy chemicals.

## 1. Introduction

Daunomycin (DM), also known as daunorubicin, is an anthracycline antibiotic widely used over the past half-century to treat various cancers such as leukemia and lymphoma [1,2,3,4,5,6,7,8,9,10,11] and is found on the WHO Model Lists of Essential Medicines [12]. Despite its useful potency for cancer therapy, the application of DM for wider use as an anticancer drug is limited due to its serious side effects, including fatal cardiac toxicity. Therefore, dissecting out the intrinsic mechanisms of action of anthracycline compounds is expected to advance the pharmaceutical development of more specific anticancer therapies while depressing their original undesirable side effects. Many studies have shown that DM acts as a topoisomerase II (Top2) inhibitor, where cell killing is caused through the generation of DNA double-strand breaks (DSBs) due to Top2 poisoning [6,7,9,10,13,14,15,16,17]. Wang et al. argued that anthracyclines interact at the interface of Top2-DNA with their sugar moieties and the cyclohexane ring A [13]. In essence, many past studies suggested that the interfacial positioning makes these anthracyclines act as molecular doorstops that prevent Top2 from re-ligating the broken strands, which ultimately results in enzyme-mediated DNA damage in the form of DSBs [10]. More recently, anthracycline compounds were found to damage chromatin due to evicting histones when intercalating [3,4,8,10]. Active studies have also been carried out to enhance the binding and anti-gene activity of daunomycin by making biochemical conjugates [1,2]. However, little work has been carried out on the relationship between the anticancer activities of anthracycline compounds and their direct effects on DNA conformation.

We previously reported that DM induces the unfolding of compact single DNA molecules and causes Top2 independent DNA fragmentation observed by fluorescence microscopy (FM) and electron microscopy [18,19]. The most widely accepted action of DM is its interaction with Top2 to form a DM–DNA–Top2 ternary complex which inhibits DNA replication and transcription processes, resulting ultimately in cell apoptosis [7,10,13,15,16,20]. In the present study, we intended to make clear the possibility of a direct effect of DM on gene expression without Top2, caused by higher-order structural changes of DNA. The effect of DM on gene expression activity was evaluated in the presence of different concentrations of spermine (SP), a natural tetravalent polyamine. Polyamines are small polycationic molecules found in all living organisms and are involved in many biological functions [21,22,23]. Through in vitro studies, it is shown that polyamines cause a significant change in the higher-order structure and biological activity of DNA [24,25,26,27,28,29,30,31]. Direct binding of polyamines to DNA and their ability to modulate DNA–protein interactions are important for the molecular mechanisms of polyamine action in cell proliferation [32]. Interestingly, the biosynthesis of polyamines in actively proliferating cells, including tumor cells, is elevated compared to normal cells [32,33,34]. Thus, we used SP to mimic living intracellular conditions of polyamines to investigate the effects of DM on gene expression versus DNA conformation.

Regarding the measurement of the gene expression, we adapted a cell-free luciferase assay using rabbit reticulocyte lysate. We also performed atomic force microscopy (AFM) observations on the conformational change of DNA by the addition of DM with and without SP.

We found that the gene expression was inhibited without Top2 and that there exists a clear correlation between the conformational change of DNA and gene expression. Through our analysis of detailed gene expression profiles, we encountered an unexpected result, namely promotion and depression of the gene expression at lower and higher concentrations of DM, respectively. We expect that our findings greatly contribute to the development of chemotherapy based on this other basic mechanism of action of DM, a representative anticancer drug.

## 2. Results

### 2.1. DM Affects Gene Expression

We examined the effect of DM on gene expression using a cell-free in vitro luciferase assay with the TnT T7 Quick Coupled Transcription/Translation System (Rabbit Reticulocyte Lysate). Protein synthesis activity as a marker of gene expression was measured using the relative luminescence intensity of the luciferin–luciferase reaction (Figure 1). Incubation (30 °C, 1.5 h) for gene expression reactions was started immediately after sample preparation was completed. Figure 1A shows how gene expression increased with rising concentrations of SP and peaked in the presence of 60 µM SP. Then, it gradually decreased with a further increase in SP concentration and was completely inhibited in the presence of SP above 200 µM. This result agrees well with our previous findings [29,31].

Based on the results shown in Figure 1A, we investigated the effects of various concentrations of DM on cell-free gene expression in the presence of three different concentrations of SP (Figure 1B); 0, 60 (the concentration where gene expression is most strongly promoted) and 400 µM (the concentration where gene expression is completely inhibited). In the absence of SP, we found that gene expression increased slightly at DM concentrations ranging from 0.1 to 1 µM and then gradually decreased with higher DM concentrations. Gene expression levels in the presence of 60 µM SP showed a similar profile of promotion and inhibition, where the promotion was more apparent. The whole profile of gene expression activity with 60 µM SP increased at low DM concentrations and then gradually decreased with rising concentrations of DM. In contrast to the experiments with 0 and 60 µM SP, gene expression was completely inhibited at 400 µM SP, regardless of the concentration of DM.

To obtain a deeper insight into the effect of DM on gene expression, we preincubated our system for various amounts of time at low temperature to equilibrate the system under conditions where gene expression is suppressed. We performed preliminary observations to check the possible effect of the pre-incubation time on the activity of gene expression and found that relatively marked changes are generated between 0 and 1.5 h of incubation. Based on this observation, we carried out experiments to check the effect of DM including the data with a much longer pre-incubation time than 1.5 h. Specifically, we incubated the reaction mixture and all the materials of cell-free gene expression (including DNA, DM and/or SP) at 0, 1.5, 3, 4.5 and 6 h in an ice-water solution (at 0 °C) before starting the gene expression reaction at 30 °C (Figure 2). Figure 2A shows the efficiency of gene expression in the absence of SP and DM at different preincubation periods. The labels at the horizontal axis indicate two times: the amount of time samples was preincubated, followed by the preincubation time plus 1.5 h of addition incubation at 30 °C for the gene expression reaction. For example, “3.0–4.5 h” indicates the luminescence intensity for samples preincubated for 3 h at 0 °C, followed by 1.5 h at 30 °C. Results were normalized to the intensity without preincubation (i.e., 0–1.5 h). Figure 2 shows the relative luminescence intensity as a marker of gene expression depending on the reaction time between DNA and various concentrations of DM in the absence (B) and in the presence (C) of 60 µM SP. The intensity was normalized to the control condition used in Figure 2A. Figure 2B, in the presence of DM without SP, shows that the gene expression activity tended to gradually decrease with the period of preincubation (Figure 2B). On the other hand, in the presence of 60 µM SP (Figure 2C), DM concentrations between 0–2 µM essentially had the same dependencies on the preincubation period, suggesting a minimum effect of DM in solutions with relatively high concentrations of SP. At higher DM concentrations (i.e., 5 and 10 µM), the suppression of gene expression by DM becomes noticeable.

### 2.2. Higher-Order Conformational Changes of DNA Caused by DM

The abovementioned observations on gene expression activity indicate a bimodal effect of DM (i.e., weak promotion and inhibition at lower and higher concentrations, respectively). To shed light on this characteristic effect of DM on gene expression, we performed AFM measurements to observe the influence of DM on the higher-order structure of DNA.

Figure 3 shows AFM images of plasmid DNA (4331 bp) in the presence of 6 µM SP (A; Control), 60 µM SP (B), 60 µM SP and 5 µM DM (C), or 60 µM SP and 10 µM DM (D). Plasmid DNA molecules tended to assemble/aggregate to each other by forming flower-like structures at 60 µM SP (Figure 3B) compared to the control (Figure 3A). Inspection of the flower-like structures indicated that neighboring DNA segments tended to align parallel to each other. Note that we detected an enhancement in gene expression efficiency at 60 µM SP, shown in Figure 1A. These tendencies correspond well with our past reports [29,31]. The addition of DM destroyed the flower-like structures by fragmenting the DNA chain, i.e., introducing DNA DSBs (Figure 3C,D).

To gain additional insights into the effect of DM on higher-order DNA structures, we used AFM on much longer DNA molecules. Single T4 GT7 DNA (166 kbp) was imaged, and representative examples of elongated coil (control, 3.5 µM SP) or flower-like (10 µM SP) conformation are shown in Figure 4A and Figure 4B, respectively. These results correspond well to AFM observations in past studies [28]. T4 GT7 DNA is ca. 40 times longer than the plasmid DNA shown in Figure 3 and is therefore expected to behave as a semi-flexible polyelectrolyte [25,26,35]. The flower-like structure was generated with a single DNA molecule at 10 µM SP. On the other hand, flower-like structures for the short circular plasmids pictured in Figure 3B usually only appeared as a result of aggregation between multiple DNAs under higher SP concentrations than that observed for the much longer T4 GT7 DNA. We used FM to further explore the effects of increasing the concentrations of SP on the higher-order structure of DNA (Appendix A). DNA was imaged in solution and showed at 10 µM SP a coexistence between a coil and a compact state, whereas at 20 µM SP we observed a fully compact state (Appendix A). This suggests a folding transition that starts before 10 µM SP and ends before 20 µM SP. We therefore chose in the following to use 10 µM SP. T4 GT7 DNA was preincubated at this concentration of SP and 10 µM DM for different time points and imaged by AFM (Figure 4C). Flower-like structures were observed following the 0 h preincubation after simple mixing with 10 µM SP and 10 µM DM (Figure 4C-i). In contrast, after a 3 h preincubation (Figure 4C-ii), the DNA became looser and was accompanied by the partial disordering of the flower-like conformation. After the 6 h preincubation (Figure 4C-iii), the flower-like structure was markedly destroyed. It is clear from our AFM observations that DM causes DNA DSBs, even in the absence of Top2. This effect of DM on DNA is consistent with earlier studies [18,19], namely FM observations of single DNA molecules in an aqueous solution which, in contrast to our AFM study, do not suffer from artifacts due to adsorption on a solid surface.

## 3. Discussion

In the present study, we showed that DM exhibits a bimodal effect and can depress/inhibit the activity of cell-free gene expression at µM concentrations. AFM observations revealed DM’s structural effects on DNA, where higher-order structural changes accompanied by fragmentation, or DSBs, on DNA occurred. Based on these findings, we would like to discuss how DM influences DNA by focusing on the relationship between the structural changes of DNA and the activity of gene expression.

To evaluate the effect of DM on gene expression, we measured changes in gene expression in the presence of different concentrations of SP (Figure 1B). Gene expression was completely inhibited in the presence of 400 µM SP, independent of the concentration of DM. Such perfect inhibition is attributable to the tightly folded and compacted state of DNA for relatively large concentrations of SP [29,31]. Polyamines can induce the conformational transition of DNA from an elongated coil into a compact globule (i.e., a coil-globule transition), accompanied by a large effective volume change on the order of 10^4^–10^5^, when the size of DNA is above several tens of kbp [24]. Thus, double-strand DNA in the compact globule state is inaccessible to RNA polymerase (blocked transcription) and DM (DNA DSBs blocked). In contrast, gene expression was weakly promoted when the DM concentration was less than 1 µM and combined with either 0 or 60 µM SP (see Figure 1B,C). Such promotion may be due to the insertion of DM into the DNA double-strand structure. Gene expression tended to be depressed when DM concentrations exceeded 1 µM. We believe the depression effect is attributable to the damaging interaction of DM and DNA observed with AFM (see Figure 3 and Figure 4), including the formation of DSBs. The time dependence on gene expression shown in Figure 2 suggests that individual DM molecules intercalate and attack certain parts of DNA molecules, inducing DSBs while remaining local. Therefore, the probability of successive attacks at other DNA locations is rather low. DM’s potential preference to position/attack locally provides important information on the actual effect of DM on genomic DNA in living cells. In relation to the weak promotion observed at lower concentrations of DM (Figure 1), Figure 2B shows that the promotive effect tends to continue but decreases with the time of incubation. On the other hand, with 60 µM SP (Figure 2C), the activity of gene expression remains at essentially the same level, irrespective of the incubation time. Although, at present, the fundamental mechanism of the promotion of gene expression at low concentrations of DM is not clear, such a specific effect of low concentration of DM on gene expression is overall consistent with a scenario of a slow DM-induced higher-order structural change of DNA and its effect on gene expression.

Relatively low concentrations of SP can promote gene expression, as shown in Figure 1A. Previous work showed that gene expression is enhanced at intermediate concentrations of polyamines (including SP) where the DNA molecule shows a shrunken conformation with an increased segment density but where the molecule is not yet too tightly packed [29,30,31]. This shrunken conformation is characterized by a flower-like structure, where enhanced parallel orientation between DNA segments is generated. In other words, the basic property of aligning DNA segments resembles that of liquid crystals, where rod-like elements prefer to be arranged in parallel even for non-condensed DNA [36,37]. However, for short DNA with several tens of bp, conformations with flower-like structure cannot occur. It is noted that short DNA behaves as a rigid rod because the persistence length and diameter of double-strand DNA are around 50 and 2 nm, respectively [38,39]. Recently, a specific effect on the promotion of gene expression was found; a longer DNA template exhibited stronger cell-free gene expression activity than a shorter one [40]. Here, we show that the length of DNA has a marked effect on the fundamental characteristics of the formation of flower-like structures (Figure 3 vs. Figure 4). Flower-like conformations with a single DNA molecule appeared when lower concentrations of SP were added to longer DNA, while shorter DNA tended to assemble with several DNA chains. Flower-like structures are shrunken conformations with a preference for locally parallel arrangements or liquid-crystal-like states of DNA segments. The higher gene expression activity for such shrunken states observed in the present study is consistent with a more general mechanism of self-regulation of genetic activity via DNA conformations [29,30,31].

We summarize the main findings of the present study, as follows: (1) DM causes the promotion and repression of gene expression, depending on its concentration. As far as we are aware, there have been no reports of the promotive effect of DM or other anthracycline antibiotics. (2) From the observation of the higher-order structure of DNA, it has become clear that the promotion of gene expression occurs when DNA is in an elongated/relaxed conformation at lower DM concentration. (3) A significant inhibitory effect on gene expression by DM is attributable to the destruction of the flower-like structure of DNA, accompanied by the generation of DSBs. Thus, the present study has revealed that DM directly affects the structure and bio-function of DNA and is expected to contribute to a deeper understanding of the anticancer activity of anthracycline antibiotics, in addition to the current insight for the inhibitory effect on Top2. Future work on the higher-order structure of genome-size DNA will be important for the development of anticancer drugs and to obtain a deeper understanding of the basic mechanisms of gene expression in living cells.

## 4. Materials and Methods

### 4.1. Materials

Daunorubicin hydrochloride (daunomycin, DM) and the antioxidant 2-mercaptoethanol (2-ME) were purchased from FUJIFILM Wako Pure Chemical Corporation (Osaka, Japan). Spermine tetrahydrochloride (SP) was purchased from Nacalai Tesque (Kyoto, Japan). The fluorescent cyanine dye YOYO-1 (1,10-(4,4,8,8-tetramethyl-4,8-diazaundecamethylene)bis[4-[(3-methylbenzo-1,3-oxazol-2-yl) methylidene]-l,4-dihydroquinolinium] tetraiodide) was purchased from Molecular Probes, Inc. (Eugene, OR, USA). T4 GT7 DNA (166 kbp with a contour length of 57 µm) and Tris-hydrochloride acid buffer (pH 7.5) were purchased from Nippon Gene (Tokyo, Japan). Plasmid DNA (Luciferase T7 Control DNA, 4331 bp) containing a firefly luciferase gene and a T7 RNA polymerase promoter sequence was purchased from Promega (Madison, WI, USA).

### 4.2. Methods

#### 4.2.1. Luciferase Assay for Gene Expression

The cell-free luciferase assay with a TnT T7 Quick Coupled Transcription/Translation System (Promega) was carried out according to the manufacturer’s instructions and previous reports [29,30,31,40]. Plasmid DNA (4331 bp) encoding a firefly luciferase gene with a T7 promoter sequence was used as the DNA template. The DNA concentration was 0.6 µM in nucleotide units. The reaction mixture containing the DNA template was incubated for 90 min at 30 °C on a Dry Thermo Unit (TAITEC, Saitama, Japan). We measured the effects of the addition of various concentrations of DM and SP on the luminescence intensity. The expression of luciferase was evaluated following the addition of luciferin as the luciferase substrate (Luciferase Assay Reagent, Promega) by detecting the emission intensity at around 565 nm using a luminometer (MICROTEC Co., Chiba, Japan). The intensity was normalized to the control condition to equal 1.0 (control measurement was performed under 90 min incubation at 30 °C in the absence of both SP and DM and without first preincubating at 0 °C).

#### 4.2.2. Atomic Force Microscopy (AFM)

AFM used an SPM-9700 (Shimadzu, Kyoto, Japan) to image and measure 0.6 µM plasmid DNA (4331 bp) or T4 GT7 DNA dissolved in 10 mM Tris-HCl buffer solution at pH 7.5 with various concentrations of DM (0–10 µM) and SP (3.5–60 µM). The DNA solution was incubated for more than 10 min and then transferred onto a freshly cleaved mica surface. Subsequently, the sample was rinsed with ultra-pure water, dried with nitrogen gas and imaged by AFM. All measurements were performed in the air using the tapping mode. The cantilever (OMCL-AC200TS-C3, Olympus, Tokyo, Japan) was 200 µm long with a spring constant of 9–20 N/m. The scanning rate was 0.4 Hz, and images were captured using the height mode in a 512 × 512-pixel format. The obtained images were plane-fitted and flattened by the computer program supplied with the imaging module.

## 5. Conclusions

In the present study, we found that DM exhibited a bimodal effect on cell-free gene expression under solution conditions optimized by the addition of SP (Figure 1). Gene expression was weakly promoted and then depressed as the concentration of DM increased until it caused complete inhibition. AFM observations on the higher-order structure of DNA indicate that flower-like conformations of DNA are formed when solution conditions promote gene expression in the presence of SP. We attribute the complete inhibition of gene expression in the reaction medium containing a large concentration of SP to the formation of a tightly compacted globule state of DNA. DM cannot disturb the DNA structure when it is in such a compact globule state. On the other hand, DM interacts with DNA when it exhibits a flower-like conformation (i.e., a relatively active genetic state), resulting in a marked effect on the higher-order structure caused by DNA damaging interactions including DSBs. The present study has revealed a clear indication that DM directly affects the structure and bio-function of DNA and is expected to contribute to a deeper understanding of the anticancer activity of anthracycline antibiotics, in addition to the current insight on the inhibitory effect on Top2.

## Figures and Tables

**Figure 1 ijms-24-06631-f001:**
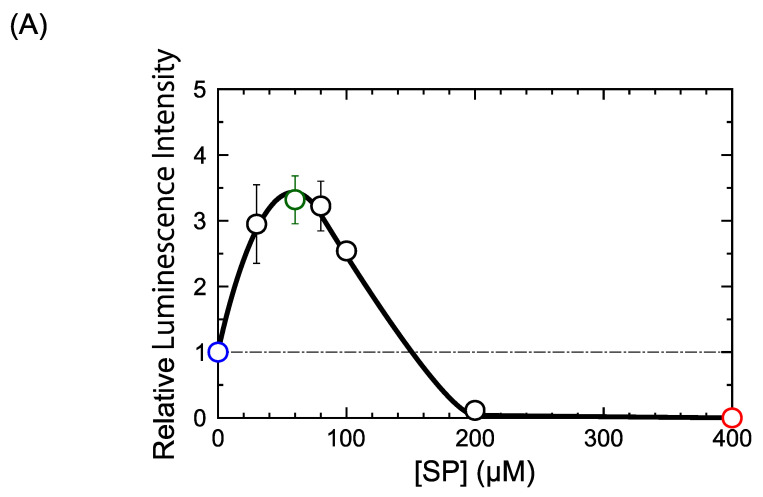
Cell-free gene expression relative to luminescence intensity. Efficiency of gene expression was measured for various concentrations of (**A**) spermine (SP) and (**B**) daunomycin (DM) at different SP concentrations. The intensity was normalized to the control (=1), where SP and DM were absent. The DNA concentration was fixed at 0.6 µM in the nucleotide unit. The error bars show the standard deviation.

**Figure 2 ijms-24-06631-f002:**
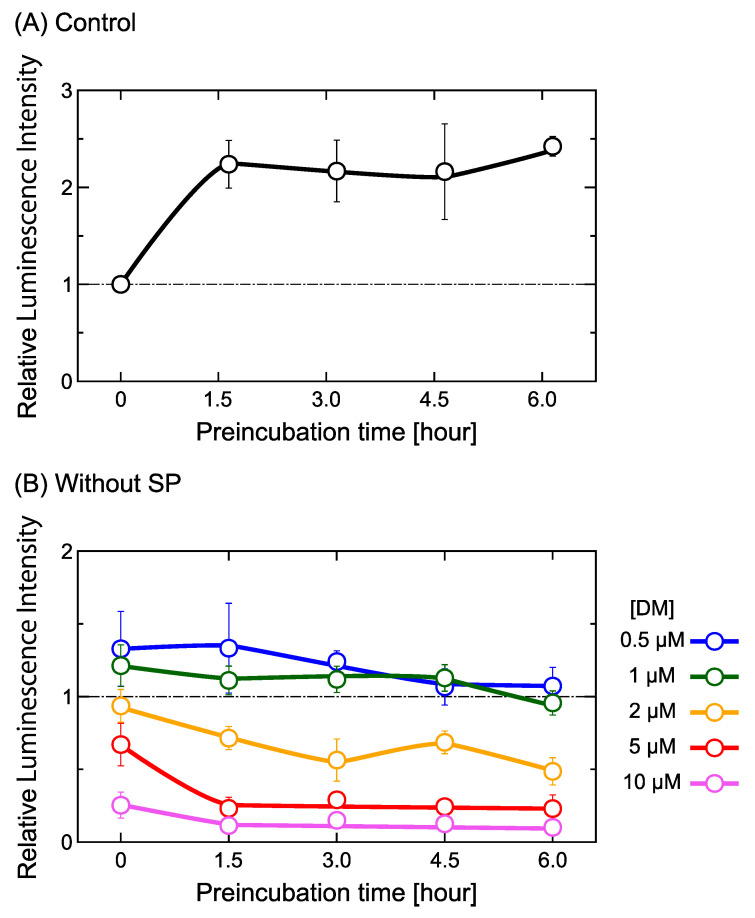
Effects of daunomycin (DM) on gene expression. Results with the same incubation period of 1.5 h at 30 °C after different preincubation periods at 0 °C are shown. The horizontal axis indicates the period of preincubation at 0 °C. (**A**) Control experiment in the absence of both spermine (SP) and DM. The intensity was normalized to the control (=1.0), as indicated at the left-end circle, where SP and DM were absent without preincubation time at 0 °C. (**B**) Gene expression for various concentrations of DM in the absence of SP. (**C**) Gene expression for various concentrations of DM in the presence of 60 µM SP. The relative luminescence intensity in (**B**,**C**) are each normalized (=1.0) with the relative luminescence intensity from (**A**) for the same incubation period. The DNA concentration was fixed at 0.6 µM in nucleotide unit. The error bars show the standard deviation.

**Figure 3 ijms-24-06631-f003:**
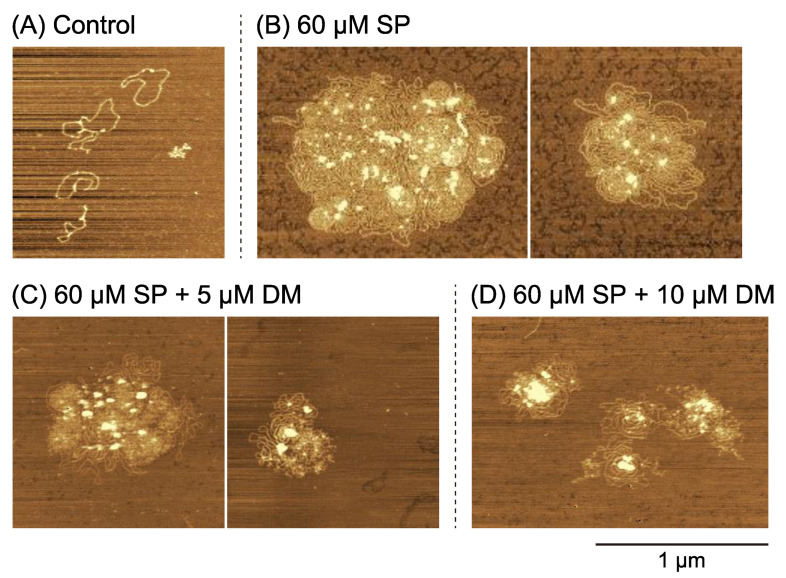
Atomic force microscopy images of plasmid DNA. Plasmid DNA (4331 bp) in the presence of various concentrations of spermine (SP) and daunomycin (DM): (**A**) control (6 µM SP), (**B**) 60 µM SP, (**C**) 60 µM SP and 5 µM DM and (**D**) 60 µM SP and 10 µM DM. The DNA concentration was fixed at 0.6 µM in nucleotide unit. The scale bar is 1 µm and applies to all subfigures.

**Figure 4 ijms-24-06631-f004:**
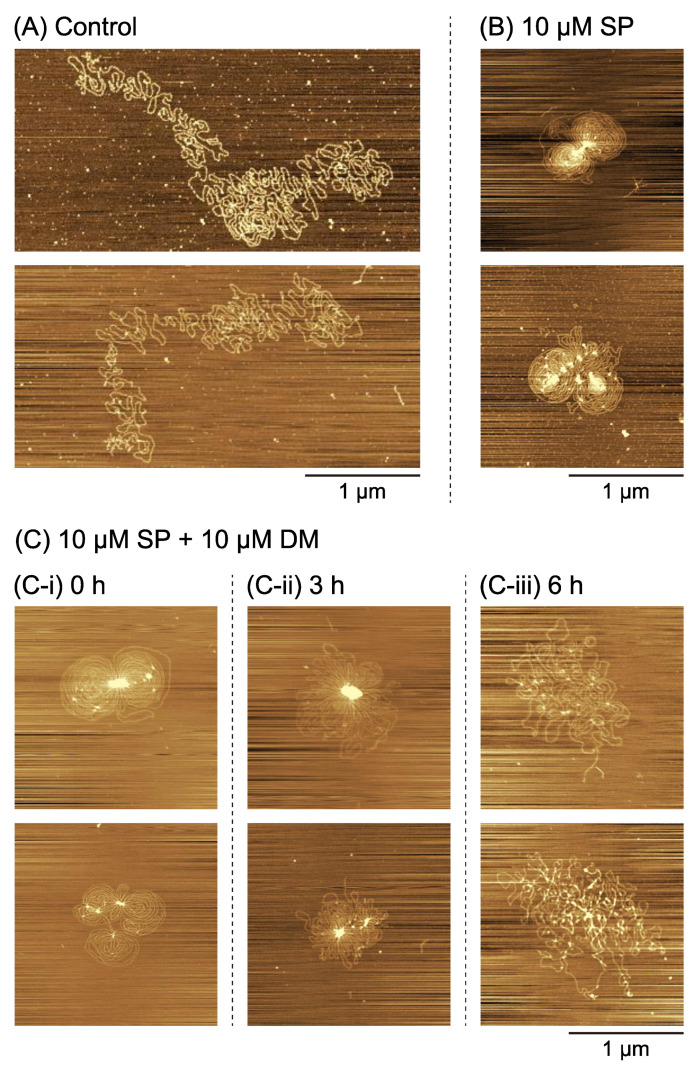
Atomic force microscopy images of T4 GT7 DNA. T4 GT7 DNA in the presence of various concentrations of spermine (SP) and daunomycin (DM): (**A**) control (3.5 µM SP) without preincubation, (**B**) 10 µM SP without preincubation; (**C**) 10 µM SP and 10 µM DM without preincubation (**C-i**), with preincubation at room temperature (24 °C) for 3 h (**C-ii**) and with preincubation at room temperature (24 °C) for 6 h (**C-iii**). The DNA concentration was fixed at 0.6 µM in nucleotide unit. The scale bar is 1 µm and applies to all subfigures.

## Data Availability

All data are contained within the article or Appendix A.

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
