# Peer review of "The Anticancer Drug Daunomycin Directly Affects Gene Expression and DNA Structure"

_ijms, 2023, doi:10.3390/ijms24076631_

Round 1

Reviewer 1 Report

In the manuscript, authors reported effect of daunomycin (DM) on gene expression and DNA structure. Here, authors highlighted at high concentration DM suppresses gene expression, causes double stranded breaks in DNA irrespective of topoisomerase II mediated inhibition. The findings are interesting however authors need to address the following before publication:

  1. While checking the effect of DM on gene expression, authors pre-incubated the sample at 0 degrees C for different time point (0-6)h and further incubated at 30 degrees C for 1.5 h to check the gene expression. Pre-incubation was performed to suppress the gene expression and to equilibrate the system. However, it would have been interesting to know the kinetics of gene expression as at lower concentration of DM, where slight activation in the expression was observed. 
  2. In Figure 1A and B was the sample pre-incubated too? At what time point was luminescence intensity measured? Also authors must include a statement on why such long pre-incubation 4.5 or 6h was performed.
  3. In Figure 2, on horizontal axis time in range (#-##) h makes it hard to follow. It would be better mention pre-incubation time as (0, 1.5, 3, 4.5 and 6)h followed by 1.5h incubation time for gene expression (which was common for all the conditions).
  4. In Figure 3, There are two images for 3B and C, are conditions same for both of them? The sizes vary so authors must include scale for each image. 
  5. Did authors check the effect of SP alone with pre-incubation at 0, 3 and 6 h time point? That result should be included in Figure 4. Also images placed adjacent to each other all controls and experimental will make the presentation easier to compare.

Reviewer 2 Report

The article is well written and easy to read. Can You explain why in Fig 1 the error bars of 60 uM SP are much larger
than at other concentrations.
It is difficult to understand what the # symbols in the title of Fig 2
represent

Reviewer 3 Report

The article “The anticancer drug daunomycin directly affects gene expression and DNA structure” by Takashi Nishio and co-authors reported the bimodal effect of the anticancer drug daunomycin on cell-free gene expression under solution conditions optimized by the addition of spermine. AFM observations on the higher-order structure of DNA showed flower-like conformations of DNA when solution conditions promote gene expression in the presence of spermine. Daunomycin showed a direct effect on the structure and bio-function of DNA.

The combined results presented here are minimal, less informative, not targeting a larger research community, and beyond the standard of high-quality results published in IJMS (Q1 Biochemistry & Molecular Biology / CiteScore - Q1). Similar investigations have been reported where daunomycin influenced DNA interactions, https://doi.org/10.4161/cbt.25328, https://doi.org/10.1093/nar/gkh527, https://doi.org/10.1021/bc2004236. Here, the authors reported the DM effect on gene expression using cell-free in vitro luciferase and conformational DNA changes caused by DM, which showed by AFM images, indicating less impact in this field. Therefore, the article is not recommended for publication in IJMS.

Round 2

Reviewer 1 Report

The revised manuscript has been improved and the concern raised were addressed. 

Reviewer 3 Report

The modified article “The anticancer drug daunomycin directly affects gene expression and DNA structure” is very general, representing no impact in this field with no novelty. Very regular work with no future impact. The effects of DM on gene expression and the conformational DNA changes caused by DM represented in this manuscript do not align with the high-quality results published in IJMS. Lack of scope and future perspective. The article is not recommended for publication in IJMS.
